# Genome-Wide Characterization of Calmodulin and Calmodulin-like Protein Gene Families in *Paulownia fortunei* and Identification of Their Potential Involvement in *Paulownia* Witches’ Broom

**DOI:** 10.3390/genes14081540

**Published:** 2023-07-27

**Authors:** Lijiao Li, Na Guo, Yabing Cao, Xiaoqiao Zhai, Guoqiang Fan

**Affiliations:** 1College of Forestry, Henan Agricultural University, Zhengzhou 450002, China; 15028032040@163.com (L.L.); guona@henau.edu.cn (N.G.); cyb201406@163.com (Y.C.); 2Institute of Paulownia, Henan Agricultural University, Zhengzhou 450002, China; 3Henan Academy of Forestry, Zhengzhou 450002, China; user7117@163.com

**Keywords:** *Paulownia fortunei*, CaMs/CMLs, gene family, phytoplasma, calcium treatment, Paulownia witches’ broom

## Abstract

As significant Ca^2+^ sensors, calmodulin (CaM) and calmodulin-like proteins (CML), have been associated with a variety of environmental conditions in plants. However, whether CaMs/CMLs are related to the stress of phytoplasma infection has not been reported in *Paulownia fortunei*. In the current study, 5 *PfCaMs* and 58 *PfCMLs* were detected through a genome-wide investigation. The number of EF-hand motifs in all PfCaMs/CMLs varied. Bioinformatics analyses, including protein characteristics, conserved domain, gene structure, *cis*-elements, evolutionary relationship, collinearity, chromosomal location, post-translation modification site, subcellular localization and expression pattern analyses, represented the conservation and divergence of *PfCaMs*/*CMLs*. Furthermore, some *PfCaMs*/*CMLs* might be involved in plants’ reaction to phytoplasma infection and exogenous calcium therapy, indicating these genes may play a role in abiotic as well as biotic stress responses. In addition, subcellular localization analysis showed that PfCML10 was located in the cell membrane and nucleus. In summary, these findings establish a stronger platform for their subsequent functional investigation in trees and further characterize their roles in Paulownia witches’ broom (PaWB) occurrence.

## 1. Introduction

Calcium (Ca^2+^), a second messenger, is crucial for the growth as well as development of plants, and is involved in responses to inner and outer stimuli [1,2,3,4]. Evidence shows that stimulations can rapidly induce an increase in intracellular Ca^2+^ concentrations, which then bind to the protein sensors and typically amplify the calcium signals [5,6], changing their behavior or capacity for interaction with subsequent proteins, and thus the signal by causing a change in protein structure [7,8,9]. The majority of Ca^2+^-binding protein sensors are EF-hand-domain-containing proteins such as calmodulins (CaMs), CaM-like proteins (CMLs), calcineurin B-like proteins (CBLs), and Ca^2+^-dependent protein kinases (CDPKs) [2,10,11].

CaMs are highly preserved in eukaryotes, which typically have four EF-hand motifs, and each EF-hand domain can bind one Ca^2+^ ion [12]. Previous studies demonstrated that in order to control the growth and development of plants and their ability to react to challenges, CaMs can bind to the target proteins and alter their function [13,14]. Among the various Ca^2+^-binding proteins, CMLs are a plant-specific class of Ca^2+^ sensors that feature one to six EF-hand motifs and share 16–75% amino acid identity with normal CaMs [12,15,16,17]. Additionally, they are distinct from CDPK and CBL members, and most CaMs/CMLs do not have any other known functional domain, with the exception of CaM7, a transcription factor that can directly control the expression of HY5 in *Arabidopsis* [18]. Different CaMs/CMLs differ in how they bind to and control target proteins, and even minor structural variations can have a big impact on how well they attach to their targets [19]. There is proof that CMLs may not be functionally redundant [17].

CaMs/CMLs are notable Ca^2+^-binding proteins sensors in higher plants, including *Vitis vinifera*, *Lotus japonicas*, *Solanum lycopersicum*, *Fragaria vesca*, *Malus domestica*, and *Triticum aestivum*, which can mediate flower development, seed germination and biotic stress tolerance [14,20,21,22,23,24,25]. For instance, the reduction function of *AtCML24* and *AtCML25* has a significant impact on pollen development, and AtCML24 can work together with ATG4b to influence autophagy growth, whereas *CML25* takes part in the Ca^2+^-mediated regulation of K^+^ influx during pollen development [26,27,28]. *AtCML39* may play a role in controlling seed germination, growth, and early establishment [29,30]. A rice-calmodulin-like gene, *OsMSR2*, can increase resistance to drought and salt [31], while *GmCaM4* can strengthen the binding ability of DNA and MYB2, and overexpression of *GmCaM4* in soybean increases its resistance to pathogens and salt stress [32]. Compared to the wild type, the *Arabidopsis cml9* mutant is more sensitive to ABA and exhibits improved salt and drought tolerance [33]. Tolerance to drought and ABA-induced stomatal closure are adversely regulated by *Arabidopsis CML20* [34]. Moreover, many previous studies have also indicated that the expression of *CaMs*/*CMLs* strongly affects the plant immune system [13]. Reduced salicylic acid levels and disease resistance are the outcomes of CaM induced binding and activation of the calmodulin-binding transcription activator CAMTA3 in *Arabidopsis* [35]. Overexpression of *AtCML43* can accelerate the hypersensitive response [36]. *AtCML9* expression is induced by pathogens (*Pseudomonas syringae*) and involved in modulating plant defense [33,37]. *AtCML8* has a beneficial effect on plant immunity and is also strongly and transiently induced by *P*. *syringae* [38]. Additionally, recent studies on the CaM/CML gene family in grapes demonstrated that the expression of *CaMs*/*CMLs* was altered under various abiotic stresses, which suggests the functional diversity in grapevine stress resistance [20]. However, up to now, *CaMs*/*CMLs* and their relationship to phytoplasma infection had yet to be found and defined in *Paulownia fortunei*. Therefore, it was vital to investigate the function of Ca^2+^ as a second messenger in the pathogenesis of PaWB disease.

*P*. *fortunei* is an important deciduous tree of *Paulownia* (Paulowniaceae), that is widely cultivated for its high adaptability and rapid growth [39]. They have high ecological, economic, and medicinal value, which make them popular among the general public [39]. However, pathogen phytoplasma can easily infect these trees, leading to the occurrence of Paulownia witches’ broom (PaWB), which has a significant negative impact on the tree’s productivity and economic benefits. To explore its role in PaWB, here, we identified the *PfCaM*/*CML* gene family in the *P*. *fortunei* genome and investigated their phylogenetics, gene structure, chromosomal locations, protein motifs, subcellular location, syntenic and promoter cis-acting elements. Moreover, the spatiotemporal expression patterns and expression profiles of healthy *P*. *fortunei* seedlings (PF), PaWB phytoplasma-infected *P*. *fortunei* seedlings (PFI), and PFI treated with rifampin (Rif) and methyl methane sulfonate (MMS) were also analyzed. Additionally, six *PfCMLs* with a response to calcium signaling under calcium treatment were examined. To better understand the Ca^2+^-mediated signal transductions in *P*. *fortunei*, as well as to aid future gene cloning and functional investigation, it would be helpful to conduct a systematic examination of the entire set of CaMs/CMLs in *P*. *fortunei*.

## 2. Materials and Methods

### 2.1. Plant Materials and Growth Conditions

All plant samples utilized in this study were from Henan Agricultural University’s Laboratory of Forestry Biotechnology in Zhengzhou, Henan Province, China. The tissue-cultured seedlings were cultured in greenhouse at 25 ± 2 °C under a 16 h day/8 h night cycle with an irradiance of 130 μmol·m^−2^·s^−1^. The apical buds, stem, and leaf from PF and PFI were collected for further study.

For calcium treatment, PF and PFI which grew approximately 1 cm root were transfered to 10 mM CaCl_2_ and normal medium (control). The apical buds were collected at 3, 6, 9, and 12 h after transformation. All samples were rapidly frozen in liquid nitrogen and kept at −80 °C for RNA extraction. For each treatment, three biological duplicates were established. *Nicotiana benthamiana* was grown in an infant incubator that had a constant temperature with 16 h/8 h light/dark.

### 2.2. Identification of the CaM and CML Family in P. fortunei

The protein sequences of *Arabidopsis thaliana* CaM and CML were used to retrieve the TAIR website (http://www.Arabidopsis.org/) in order to determine the CaM and CML proteins in *P*. *fortunei*. The detailed *Arabidopsis thaliana* genes used for identification are listed in Appendix A. These sequences served as search criteria for the *P*. *fortunei* genome’s candidate members using BLASTP (E cut-off value < 1 × 10^−5^). Coding sequences without ATG were considered partial and defective, which were eliminated. Then these non-redundancy sequences were verified using SMART (http://smart.embl-heidelberg.de/) and NCBI-CDD (http://www.ncbi.nlm.nih.gov/cdd) programs. Sequences possessing only the EF-hand domains were considered to be CaMs/CMLs, similarity greater than 90 to AtCaM2 was considered to be CaMs.

Protein theoretical isoelectric point (pI) and molecular weight (MW) prediction were performed in Expasy ProtParam (http://web.expasy.org/protparam/). Subcellular location prediction was used five tools (loctree3, https://rostlab.org/services/loctree3/; Yloc, https://abi-services.cs.uni-tuebingen.de/yloc/webloc.cgi; WoLF PSORT, https://wolfpsort.hgc.jp/; Plant-PLoc, http://www.csbio.sjtu.edu.cn/bioinf/plant/; ProtComp, http://linux1.softberry.com/berry.phtml/). Using NetPhos 3.1 (http://www.cbs.dtu.dk/services/NetPhos/), Phosphorylation site prediction was performed on the serine, threonine, and tyrosine residues, and the score threshold was settled at 0.8. Methylation, acetylation and ubiquitination site were predicted on website (http://msp.biocuckoo.org/) with high threshold.

### 2.3. Chromosomal Locations, Motifs and Gene Structure Analyses

Based on *P*. *fortunei* genome databases, chromosomal locations and corresponding sequence of *PfCaMs*/*CMLs* were retrieved. “Gene Location Visualize (advanced)” program of TBtools was used for gene mapping (v.1.09868) [40]. The exon-intron structure of *PfCaM*/*CML* genes was conducted using the GSDS2.0 (Gene structure display server; http://gsds.gao-lab.org/) program with default settings. The MEME suite (http://meme-suite.org/tools/meme) was used to identify preserved domains. Normal search settings were a maximum of 10 motifs and an ideal motif width between 6 and 50.

### 2.4. Phylogenetic and Duplication Analyses

The phylogenetic trees, including *P*. *fortunei*, *Malus domestica* (4 CaMs and 58 CMLs), and *Arabidopsis thaliana* (7 CaMs and 50 CMLs), were constructed via MEGA with Maximum Likelihood (ML) approach with the default settings, and bootstrap analysis using 1000 replicates was performed [24,41]. Collinearity analysis of *A*. *thaliana and P*. *fortunei* was obtained using MCScanX, and they were represented using Circos in TBtools software (v.1.09868) [41]. The nonsynonymous substitution rate (Ka), the synonymous substitution rate (Ks), and Ka/Ks ratio of *PfCaMs*/*CMLs* were analyzed using TBtools (v.1.09868).

### 2.5. Cis-Acting Element Analysis

As previously reported, the promoter of PfCaMs/CMLs was a 2000 bp stretch upstream of the start codon [22,25]. The cis-acting elements of PfCaMs/CMLs were discovered through PlantCARE (http://bioinformatics.psb.ugent.be/webtools/plantcare/html/) [42].

### 2.6. Gene Expression Analysis

RNA-seq data of PF, PFI, and PFI under MMS or Rif treatment at different concentrations or time points were downloaded from NCBI (access number are SRS6658602, SRS6658613, SRS6658617, SRS6658616, SRS6658600, SRS6658592, SRS6658601, SRS6658603, SRS6658608, SRS6658607), as well as the transcriptome data of leaf, trunk phloem and trunk cambium from 4-year-old *P*. *fortunei* (access number are SRS6658596, SRS6658598, SRS6658597) [39]. Heatmaps constructed using the TBtools based on *PfCaMs*/*CMLs* gene expression levels [41].

For the gene expression verification, total RNA was isolated using the TIANGEN Simple Total RNA kit (Beijing, China). For each sample, 1 μg of total RNA was reverse-transcribed into total DNAs using the GeneStar starscript III All-in-one RT, and 2×RealStar Green Fast Mixture with ROX II (Beijing, China) were used to Real-Time qPCR. *PfActin* as an internal control, and relative expression levels were calculated according to the 2^−∆∆CT^ (cycle threshold) method. The Appendix A lists the primer sequences utilized in the qRT-PCR analysis.

### 2.7. Vector Construction and Subcellular Localization Analysis

Utilizing the ClonExpress II One Step Cloning Kit from Vazyme, the entire PfCML10 coding sequence was cloned into the pSAK277-eGFP at EcoRI restriction sites for subcellular localization study. The recombinant plasmids *35S::PfCML10-eGFP* were transferred into *Agrobacterium tumefaciens* strain GV3101 for *Nicotiana benthamiana* leaf transient expression. Then, these transformed *N*. *benthamiana* were cultured at 25 °C for 72 h, using confocal laser scanning microscope (Zeiss LSM 710, Göttingen, Germany), and eGFP signals were seen and captured using camera. The primer sequences are listed in Appendix A.

### 2.8. Statistical Analysis

The data were analyzed, and all statistical tests were run using SPSS 19.0 (Chicago, IL, USA). Asterisks denote statistical significance at various levels (* *p* < 0.05 and ** *p* < 0.01), and values mean SD.

## 3. Results

### 3.1. Identification Analysis of CaM/CML Family Genes in P. fortunei

Seven AtCaM and fifty AtCML proteins were used as query sequences to find the CaM and CML genes in the *P*. *fortunei* genome. Finally, the *P*. *fortunei* genome contained 58 PfCMLs and a total of 5 PfCaMs. Based on the chromosomal positions, the *PfCaM*/*CML* genes were called sequentially consecutively from *PfCaM1* to *PfCaM5* and *PfCML1* to *PfCML58*. All PfCaMs had four EF-hands, their protein length (aa), isoelectric points (pI), theoretical molecular weights (MW) ranged from 149 (PfCaM1/2/5) to 188 (PfCaM4) aa, 4.07 (PfCaM3) to 4.8 (PfCaM4), and 16.85 (PfCaM1/2/5) to 21.78 (PfCaM4) kDa, respectively (Table 1). With one to four EF-hands, PfCMLs differed more from PfCaMs in terms of their features. PfCML proteins ranged in length from 84 (PfCML9/30) to 275 (PfCML51) AA residues, their molecular weights ranged from 9.20 (PfCML30) to 31.07 (PfCML7) KDa, and pI varied from 3.67 (PfCML27) to 8.94 (PfCML51), respectively (Table 1). The other information for the PfCaM/CML proteins, including functional feature sites and methionine percentage, is shown in Table 1.

We constructed a phylogenetic tree using the 63 PfCaMs/CMLs, 62 MdCaMs/CMLs, and 57 AtCaMs/CMLs to examine the evolutionary relationship. According to the findings, there are nine groups that can be made up of these 63 PfCaMs/CMLs. (Figure 1), comparable to those identified in *A*. *thaliana*, *Malus domestica*, and *Hordeum vulgare* [24,41,43]. PfCaMs were placed in group V, and PfCMLs were assigned to all of the groups. Group VI had the most PfCaM/CML members (11), followed by groups IX (10) and II (9), whereas groups I (5) and III (4) had few genes. Additionally, the dendrogram showed that there was no clear single homolog CaMs/CMLs among *P*. *fortunei*, *Malus domestica* and *A*. *thaliana*, suggesting that CaMs/CMLs from *P*. *fortunei*, *A*. *thaliana*, and *Malus domestica* have obvious evolutionary differences.

### 3.2. Conserved Motifs and Gene Structure of PfCaMs/CMLs

Phylogeny, conserved motifs, and gene structures of *PfCaMs*/*CMLs* were further analyzed (Figure 2). Family members with close evolutionary ties have substantial sequence similarity and similar motifs in general (Figure 2A). A total of 10 domains were identified and confirmed via the MEME tool. According to the findings, PfCaM/CML motifs were shown to be diverse among PfCMLs but relatively conserved among PfCaMs (Figure 2B); more information about conserved motifs is provided in Appendix A. Furthermore, the existence of the same type of conserved domains may indicate functional similarities in the PfCaM/CML families. In addition, the gene structures were analyzed to examine the structural diversity of *PfCaM*/*CML* genes (Figure 2C). Members from groups II, IV, and V were intron-rich, while members from the three groups I, III, and VI were intron-less. Notedly, groups VII, VIII, and IX contained no intron structure. The phylogenetically closely related genes in other plant species shared the same or a very comparable exon-intron structure, although PfCaMs/CMLs in various groups exhibited varied exon-intron structural characteristics.

### 3.3. Chromosomal Location and Duplication Analysis

To identify the distribution of *PfCaMs*/*CMLs* in *P*. *fortunei* chromosomes, we examined their chromosomal position. Of the 63 *PfCaMs*/*CMLs*, 53 genes were distributed throughout all of the *P*. *fortunei* chromosomes, except chromosome 4 (Figure 3), and the distribution appeared to be uneven. Chromosome 11 contained the most *PfCML* genes (7 members). Chromosome 13 comes next, which had six *PfCML* genes. Chromosomes 7 and 6 contained four genes. Chromosomes 3, 5, 12, 15, and 19, each had three *PfCMLs*. In addition, there is only one *PfCML* gene that was simultaneously distributed on chromosomes 9, 18, and 20. To understand the evolution of *PfCaMs*/*CMLs*, collinearity diagrams among *PfCaMs*/*CMLs* were analyzed. The findings demonstrated that the *P*. *fortunei* genome exhibited gene duplication in some *PfCaMs*/*CMLs* (Figure 3, Appendix A). The majority of *PfCaMs*/*CMLs* included 1–3 paralogous genes in *P*. *fortunei*, while PfCML11 had four paralogous genes in *P*. *fortunei*. The *Ka*/*Ks* ratios of gene pairs were <1.0, indicating that these *PfCaM*/*CML* pairs were subjected to purifying selection following duplication (Appendix A). We also carried out a synteny study of the *CaM*/*CML* genes in *P*. *fortunei* and *A*. *thaliana*. Thus, 50 orthologous pairs were found in *P*. *fortunei* and *A*. *thaliana* (Appendix A). Eight genes (*PfCML10*/*16*/*20*/*22*/*31*/*47*/*54*/*56*) had two orthologous genes in *A*. *thaliana*, while three genes (*PfCML11*/*12*/*48*) had three orthologous genes in *A*. *thaliana*. However, less than 50% identity existed in around one-third of the orthologous pairs, indicating that the function of gene pairs may have been differentiated. According to some significant percentages of identity between PfCaMs/CMLs and AtCaMs/CMLs, PfCaM/CML protein functions and sequences may be substantially conserved.

### 3.4. Diversified Expression Patterns of PfCaMs/CMLs

We employed transcriptome sequencing data and thoroughly examined these expression levels to investigate the expression patterns of *PfCaMs*/*CMLs* in sapling trees. In total, 57 *PfCaMs*/*CMLs* are expressed in trunk phloem, leaf, and trunk cambium (Figure 4A). Compared with other genes, five genes (*PfCaM4*, *PfCML8*/*17*/*32*/*36*) were highly expressed in trunk phloem. In terms of the leaf transcriptome, PfCML27 had the greatest expression level, next to *PfCML8*/*4*/*46*/*16*. Particularly, the expression level of *PfCML27* was almost 2-fold higher than that of *PfCML8*. Six genes (*PfCML48*/*20*/*8*/*17*/*49*/*36*) appeared to have a relatively high expression in trunk cambium, which indicated that their function may relate to secondary growth. Notably, *PfCML8* was highly expressed in all three tissues, which implied that *PfCML8* is positively involved in developmental processes of trunk phloem, leaf, and trunk cambium. The discovery suggested that *PfCaMs*/*CMLs* may be engaged in a variety of biological processes throughout the growth and development of *P*. *fortunei*.

Additionally, expression analysis of *PfCaM*/*CML* genes that responded to PaWB phytoplasmas infection was performed. Compared with PF, expression of 35 *PfCaMs*/*CMLs* was up-regulated, and 17 *PfCaMs*/*CMLs* were down-regulated in PFI. Among them, 15 genes (*PfCML10*/*13*/*15*/*20*/*22*/*26*/*27*/*28*/*29*/*31*/*33*/*34*/*43*/*44*/*47)* were up-regulated significantly in the PFI (fold change > 2), while 4 genes (*PfCML38*/*42*/*49*/*58*) were observably down-regulated (fold change > 2) (Figure 4B,C). After MMS or Rif treatment, the relative expression levels of *PfCaMs*/*CMLs* were changed, which met the trend of phytoplasma concentration, and qRT-PCR results further confirmed the expression patterns (Figure 4D). Based on these, our data suggested that *PfCaMs*/*CMLs* may be associated with PaWB occurrence.

### 3.5. Analysis of the Cis-Elements in the Promoter Region of the PfCaMs/CMLs

The coordinated regulation of stress-induced gene expression in stressful situations was carried out via transcriptional networks. We assumed that PfCaMs/CMLs would be targets of stress-regulated transcription factors and may be downstream of the transcriptional network regulating stress resistance. In light of this, we investigated the *cis*-elements associated with plant growth, hormones, and stress responses in the promoter region of the *PfCaMs*/*CMLs*. As shown in Figure 5, the mainly *cis*-acting elements of *PfCaMs*/*CMLs* were grouped into four types. *Cis*-acting elements are associated with responses to growth, such as meristem expression (CAT-box) and endosperm expression (GCN4-motif); phytohormones, such as abscisic acid (ABRE), salicylic acid (TCA-element), gibberellin (p-box), methyl jasmonate (CGTCA-motif), ethylene (ERE) and auxin (TGA-element and AUXRR-core); adversity, such as anoxia stress (ARE), low temperature (LTR), drought (MBS); defense, and stress-related elements (TC-rich repeats/WUN-motif) distributed in the promoter regions of the *PfCaMs*/*CMLs*. The fact that *PfCaMs*/*CMLs* had the same or distinct cis-acting areas suggests that these genes can occasionally be regulated simultaneously in response to stress or a particular external stimulus. Therefore, our data indicated that *PfCaMs*/*CMLs* should respond to *P*. *fortunei* stress stimulus.

### 3.6. Post-Translation Modification and Subcellular Localization Prediction

To uncover the relative regulatory mechanisms of PfCaMs/CMLs, we predicted post-translational modification (PTM) sites, such as methylation, acetylation, ubiquitization, and phosphorylation (Figure 6). We found up to 22 potential phosphorylation sites in PfCML52, with at least 1 putative phosphorylation site found in PfCaMs/CMLs (Appendix A). Moreover, seven and six CaMs/CMLs were found putatively at methylation sites and ubiquitization sites. More than 50% of the family members contained acetylation sites, and three members had up to six acetylation sites (Figure 6, Appendix A). These findings laid the groundwork for further research into how the PTM affects the structure and roles of PfCaM/CML proteins.

The potential subcellular localization of the PfCaM/CML proteins was predicted, as shown in Appendix A. Except for PfCML23 and PfCML37, which were located outside the cell, most PfCaMs/CMLs were located at extranuclear sites, such as cytoplasm, vacuole, and chloroplasts. To further substantiate the subcellular localization of the PfCaM/CML proteins, we produced a temporary expression construct (35S::PfCML10-GFP). The PfCML10-GFP fluorescence signal was found in the cytoplasm and nucleus, while the GFP-free control signal was present throughout the entire population of cells (Figure 7).

### 3.7. Expression Analysis of PfCMLs in Response to Calcium Signaling

To explore the potential regulation of *PfCaM*/*CML* genes in response to calcium signaling, the expression levels of six candidate *PfCMLs* (*PfCML10*/*15*/*28*/*34*/*38*/*57*) that were selected according to the RNA-seq result were analyzed under a high concentration of calcium. Six *PfCMLs*’ expression levels changed in response to calcium, and there were differences in the calcium response time between PF and PFI (Figure 8). Among them, the expression levels of four genes (*PfCML10*/*15*/*34*/*38*) and *PfCML28* increased within 6 h and 9 h after the calcium treatment in PFI, respectively. Nevertheless, three genes (*PfCML15*/*34*/*57*) increased to the highest expression level at 9 h in PF, and *PfCML57* revealed the greatest expression levels, showing 50-times greater expression at 9 h in the CaCl_2_ treatment than control. These results suggested that the expression patterns were modified by increased Ca^2+^ concentration., and six *PfCMLs* should be variously involved in PaWB occurrence.

## 4. Discussion

Ca^2+^ is a crucial second messenger that performs a key role in plants [6]. A plethora of Ca^2+^ binding proteins that act as Ca^2+^ sensors decipher complicated Ca^2+^ signals [44]. Ca^2+^ binding proteins react to the fluctuating intracellular Ca^2+^ level and boost later signaling, triggering a physiological reaction to the signal. Among Ca^2+^ sensors, CaMs and CMLs are the most universally conserved major calcium sensor proteins in plants. However, no report on participation of plant *CaMs*/*CMLs* in *P*. *fortunei* has been found. Additionally, most *CaM*/*CML* investigations have mostly focused on herbaceous plants, particularly *A*. *thaliana* and *Oryza sativa* [26,27,28,29,30,31,33,38], and have ignored this family in angiosperm deciduous tree species. We completely characterized the CaM/CML families in *P*. *fortunei* in the current work, including the phylogeny, gene structures, conserved motifs, gene duplications, synteny analysis, expression patterns, *cis*-acting elements, subcellular localization, and PTM sites. Additionally, six *PfCMLs* were proposed to be involved in PaWB occurrence. These findings offer a whole comprehension of the *CaM*/*CML* gene families in deciduous tree species within the ngiosperm and establish important groundwork for subsequent research on how they affect tree growth and development.

According to our investigation, we discovered 58 *CML* genes in *P*. *fortunei* (Table 1), somewhat more than those in *A*. *thaliana* (50 *CML*-encoding genes), even if certain clusters of gene duplication were visible. The total amount of members of the PfCaM/CML gene families falls short of what we anticipated when we took into account the ratio of 3.0 putative *P*. *fortunei* genome size for *A*. *thaliana*. At least two substantial genome-wide duplications were performed on the *P*. *fortunei* genome [39], which could lead to numerous imperfect and incomplete coding sequences. Moreover, prior research has shown that numerous *P*. *fortunei* chromosomal pairs have the same fusion pattern and that substantial gene loss happened in an asymmetrical and reciprocal manner [39]. As a result, many fewer *PfCaM*/*CML* genes than we anticipated were found in *P*. *fortunei*. Nevertheless, approximately 84% (53 of 63) of the *PfCaM*/*CML* genes were linked to duplications, suggesting these genes were functionally redundant. According to our analysis of selective pressure, purifying selection was the dominant force during the expansion of the *CaM*/*CML* gene families in *P*. *fortunei*, implying that these duplicate genes may still maintain their original functions.

The majority of *PfCaMs*/*CMLs* from the same group shared comparable gene structures and conserved domains (Figure 2), as well as the subcellular localization prediction (Appendix A). *PfCaMs*/*CMLs* were classified into nine groups, consistent with *A*. *thaliana* [41]. Moreover, the intron-exon structural analysis revealed that the majority of *PfCML* genes lack an intron (Figure 2). This intron-exon structure may have already existed in common ancestor genes [43,45]. Despite not contributing to protein sequences, introns’ relative placements provide some hints for forecasting how genes and the proteins they encode change, which helps the gene family become more diverse [46,47]. Therefore, our data revealed that CaM/CML families shared a conserved gene structure, supporting the classification and evolutionary relationships in *P*. *fortunei*. This evidence also suggested plant CaMs/CMLs belonging to the same subgroup might play roles that are comparatively preserved. However, the protein composition and quantity of EF-hand domains in CaM/CML members, which are associated with PTM sites and Ca^2+^-binding characteristics, determine how well they operate. In actuality, different plant species have different numbers of EF-hand patterns. Two to six EF-hand motifs are generally present in *A*. *thaliana* CML proteins [41], but PfCMLs contained one to four preserved EF-hand motifs. In addition, most PTM sites were also different in the same group. Combined with collinearity analysis, these data suggested that the function of partial gene pairs may have been differentiated.

The patterns of gene expression and function were closely connected. According to RNA-seq and qRT-PCR analyses, the genes in the PfCaM/CML families have a variety of expression patterns (Figure 4). Previous studies indicated that MMS and Rif were confirmed to significantly inhibit PaWB in two different ways, and PFI treated with certain concentrations of MMS or Rif could transform healthy morphology; PaWB phytoplasma was undetectable in apical buds after 20 days of treatment [39]. Based on RNA-Seq of *PfCaMs*/*CMLs*, almost all *PfCaMs*/*CMLs* displayed higher transcript levels in buds of PFI, except for *PfCML38*, for which the expression level was down-regulated in PFI. Even so, these gene expression patterns were also positively correlated with detectable levels of PaWB phytoplasma. These findings suggest that *PfCaM*/*CML* genes should interact with or function in the PaWB. For this purpose, the stress-related regulatory elements in the promoter regions were examined. Numerous cis-elements related to hormones and stress reaction were present in almost all members (Figure 5), which have been associated with biotic stress. Furthermore, the PfCML10 was located in the cell membrane and nucleus within *N*. *benthamiana* cells, similar to that in *Arabidopsis* AtCML24, modulating the actin cytoskeleton; then, it regulates pollen tube growth and affects autophagic progression [27,28]. The involvement of *PfCML10* in the autophagic process during phytoplasma infection is assumed, such as degradation of chlorophyll and chlorophyll–protein complexes. Presumably, these proteins may be signaling proteins, which possibly contribute to the occurrence of PaWB. Co-immunoprecipitation, yeast two-hybrid, and proteomics techniques could be used to identify the target proteins of PfCML10.

The variation in the expression of *PfCaMs*/*CMLs* in response to calcium treatment may be relevant in examining its functional significance in plant growth and development, considering that Ca^2+^ plays a significant part in plant stress response. We analyzed the expression of six *PfCMLs* in apical buds via qRT-PCR. In PFI, these six selected *PfCMLs* were up-regulated after CaCl_2_ treatment in comparison with PF, indicating that Ca^2+^-permeable channels may be activated by high concentrations of exogenous calcium ion, increasing the intracellular Ca^2+^ concentration, thus generating signal transduction.

By reducing BRI1-mediated reactions, *AtCML8* could positively control defense systems, enabling full PTI establishment [48]. The expression level of *PfCML15*/*28*, which is closely related to AtCML8 in evolution and high homology (43.7% and 64.5%), were significantly up-regulated in PFI. In this scenario, the significant up-regulation of *PfCML15*/*28* might be connected to the efficient suppression of *BRI1* in *P*. *fortunei*. Ma et al. [49] found *AtCML24* favorably regulates defense resistance to the bacterial pathogen *P*. *syringae*. As a collinear gene of *AtCML24*, PfCML10 shares more than 50% identity with AtCML24 and, in response to phytoplasma infection, these indicated that it might be involved in regulating resistance to pathogens. However, in view of the short treatment time of calcium ions, we did not notice a change in the growth and development of PFI. Whether the PaWB occurrence is mediated by these *PfCaMs*/*CMLs* needs to be further investigated in the transgenic *P*. *fortunei*. It will be possible to learn more about the genetic functions and underlying molecular mechanisms of these *P*. *fortunei* transgenic plants through subsequent efforts to obtain mutants and overexpression lines.

## 5. Conclusions

We found and described 5 PfCaM and 58 PfCML proteins with functional EF-hand domains in the *P*. *fortunei* genome. According to expression studies, PfCaM/CML members exhibit a variety of expression patterns in various tissues and stress responses, and their expression levels follow the trend of phytoplasma concentration. Meanwhile, partial *PfCaM*/*CML* members’ response to calcium stimulus, and six *PfCMLs* were selected that were suggested to be possibly differently involved in PaWB occurrence. To date, it is yet unknown how PfCaMs and CMLs work in *P*. *fortunei*. Thus, in order to clearly identify the functions of these *PfCaM*/*CML* genes in the occurrence of PaWB, we next examined the phenotypes of the CRISPR/Cas9-targeted PfCaMs/CMLs knockout *P*. *fortunei* trees. These results pave the way for further investigations into the role of PfCaMs/CMLs in calcium signaling and stress responses during PaWB phytoplasma infection.

## Figures and Tables

**Figure 1 genes-14-01540-f001:**
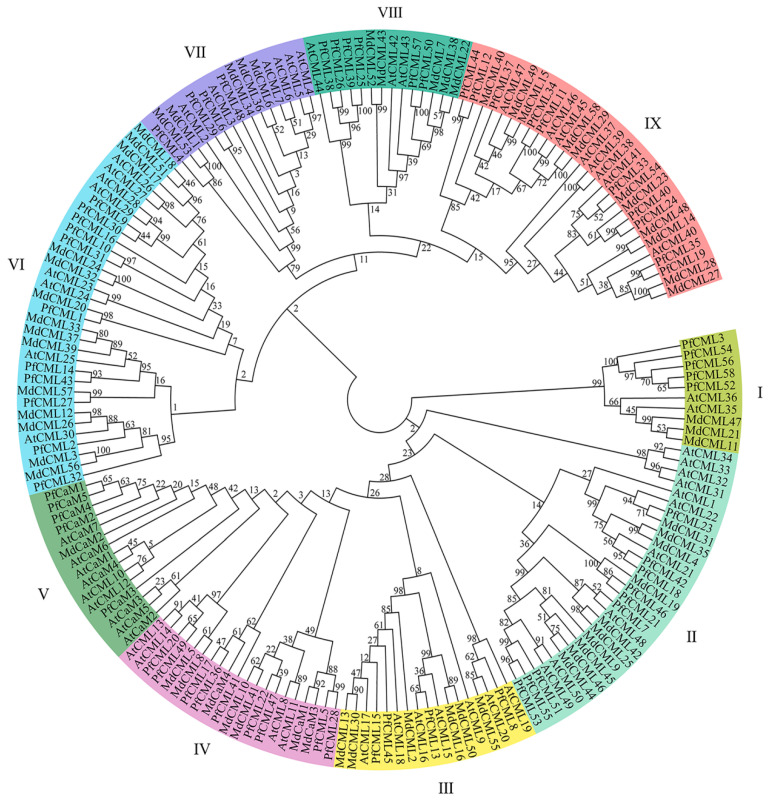
The phylogenetic tree of CaM/CML from *Paulownia fortunei*, *Arabidopsis thaliana*, and *Malus domestica*. Phylogenetic distance calculate by MEGA X program and the Maximum Likelihood (ML) method, using bootstrap values of 1000. Based on the protein sequence similarities between *Paulownia fortunei*, *Arabidopsis thaliana*, and *Malus domestica*, the members were divided into 9 groups (I–IX).

**Figure 2 genes-14-01540-f002:**
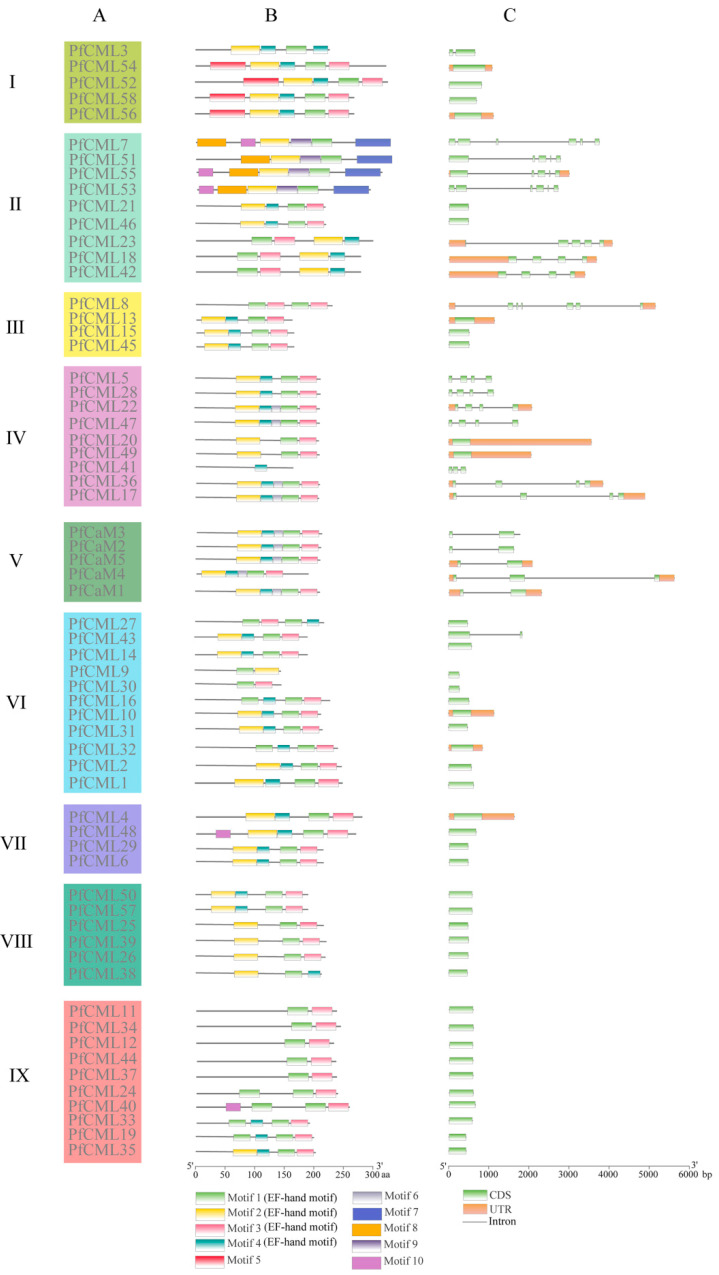
Phylogenetic groups, motif compositions and gene structures of *PfCaMs*/*CMLs*. (**A**) Phylogenetic classification of PfCaM/CML proteins; (**B**) MEME analysis revealed a schematic representation of the conserved motifs among the PfCaM/CML proteins. Each color stood for a distinct motif; (**C**) Gene structures of *PfCaMs*/*CMLs*. The CDS, UTR and Intron were appeared by green boxes, yellow boxes, and grey line, respectively.

**Figure 3 genes-14-01540-f003:**
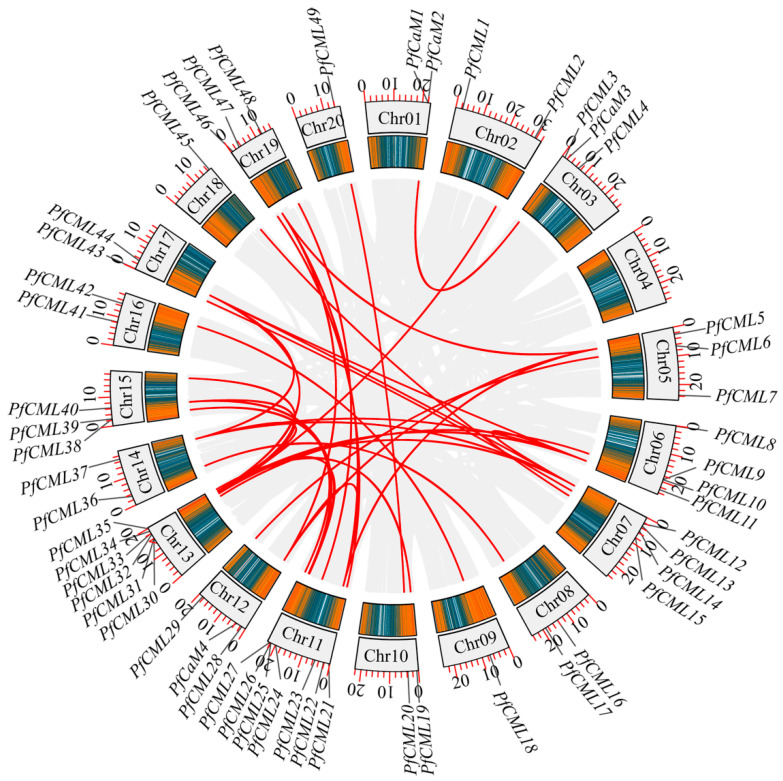
Chromosomal location and collinearity analysis of *PfCaMs*/*CMLs* in the *Paulownia fortunei* genome. The relative positions of the 53 *PfCaMs*/*CMLs* marked on chromosomes (Chr01–20). Red line represented the collinearity gene pair in the *P*. *fortunei* genome. The coloured boxes represented gene density, the more orange the colour, and the higher the density.

**Figure 4 genes-14-01540-f004:**
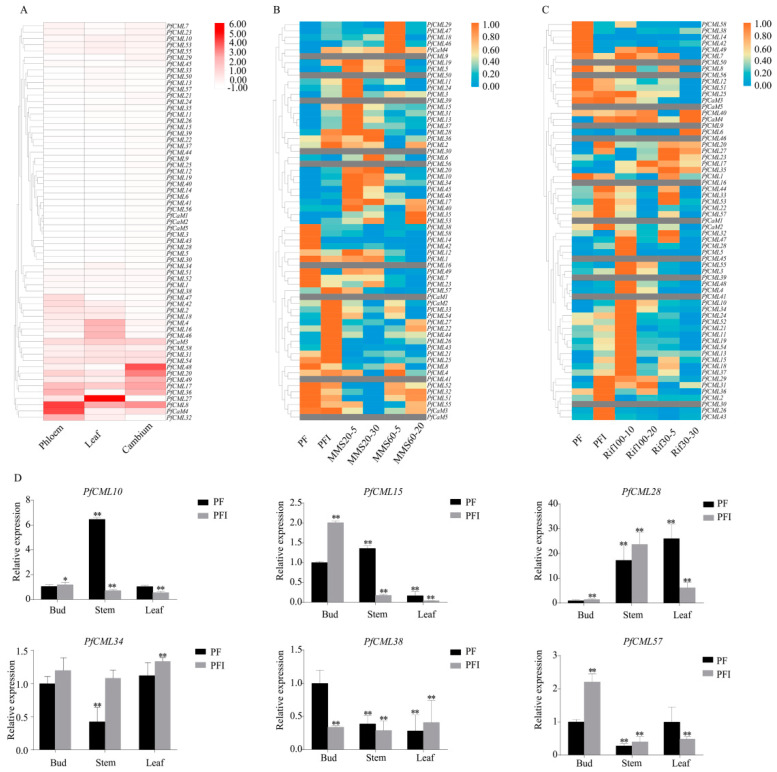
Diversified expression patterns of *PfCaMs*/*CMLs* genes. (**A**) Heatmap of *PfCaMs*/*CMLs* genes in 4-year-old *Paulownia fortunei*. The more red the color the higher the expression. PF-phloem, PF-leave and PF-cambium represent the phloem, leaf and cambium of *Paulownia fortunei*, respectively; (**B**) Heatmap of *PfCaMs*/*CMLs* genes expression in response to MMS in PFI. MMS20-5/30 and MMS60-5/20 were expressed that PFI was treated with 20 mg·L^−1^ MMS for 5/30 d, and 60 mg·L^−1^ for 5/20 d, respectively; PF, *Paulownia fortunei* seedlings; PFI, Paulownia witches’ broom (PaWB) phytoplasma-infected *Paulownia fortunei* seedlings. (**C**) Heatmap of *PfCaMs*/*CMLs* genes expression in response to Rif in PFI. Rif30-5/30 and Rif100-10/20 were expressed that PFI were treated Rif with 30 mg·L^−1^ for 5/30 d and 100 mg·L^−1^ for 10/20 d, respectively. High expression levels were shown in orange, and low expression levels were shown in blue; (**D**) qRT-PCR of *PfCaMs*/*CMLs* genes expression in apical bud, stem and leaf in PF and PFI. Error bars represented SD values from biological repeats, and significance testing constructed by one-way ANOVA (** *p* < 0.01, * *p* < 0.05).

**Figure 5 genes-14-01540-f005:**
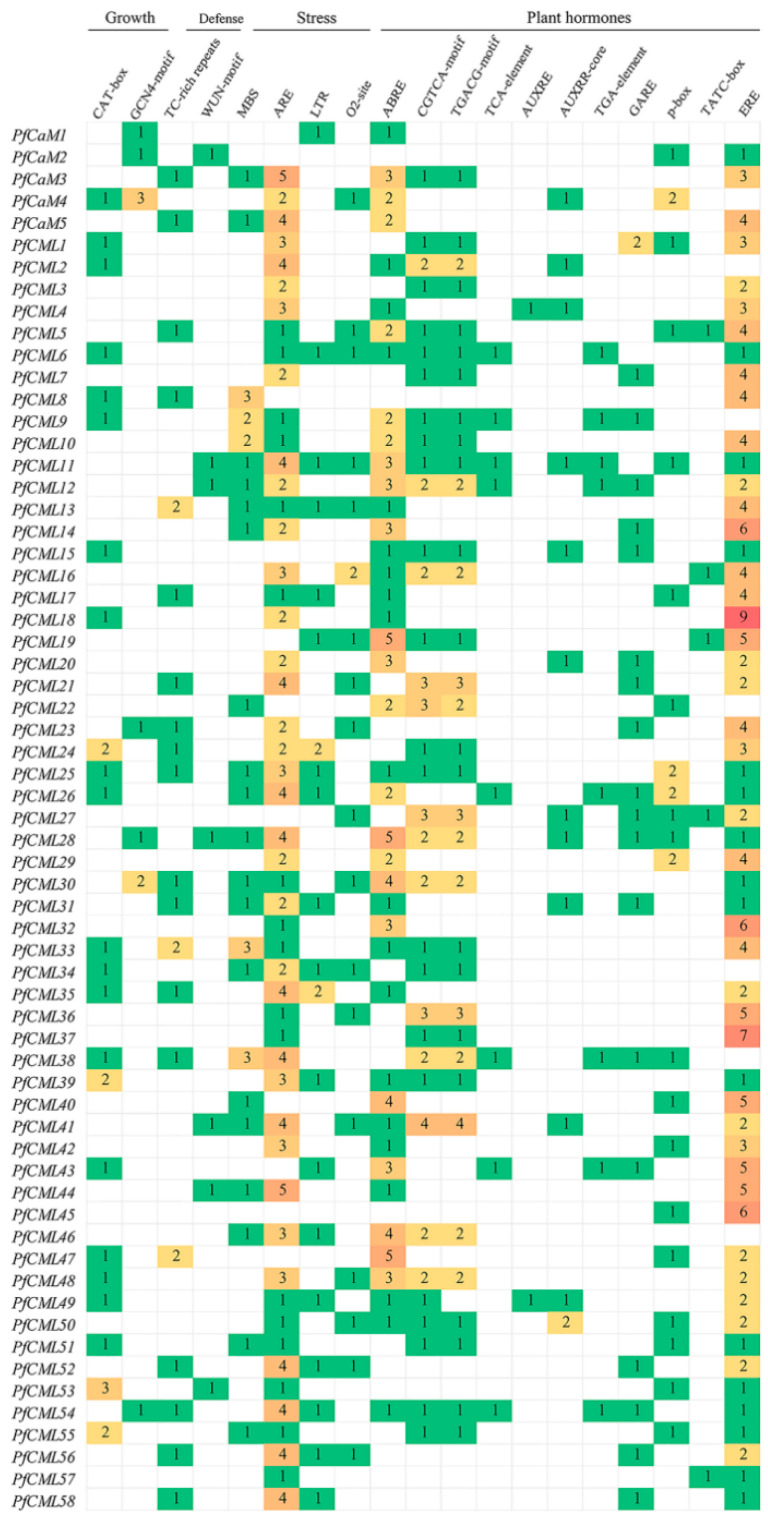
Representatives of the major *cis*-regulatory elements identified from *PfCaMs*/*CMLs* promotors. PlantCARE was used to determine the *cis*-acting elements of *PfCaMs*/*CMLs*. The *cis*-acting elements were presented in four types (Growth, Defense, Stress, and Plant hormones). The numbers represented amount of *cis*-acting elements, and the blank space meant the gene didn’t have the regulatory element. The different colours also represented the differences with the number of elements.

**Figure 6 genes-14-01540-f006:**
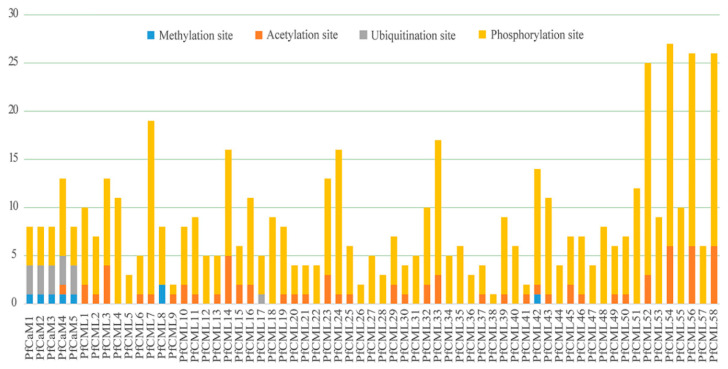
Schematic representation of four potential post-translation modification of *Paulownia fortunei* CaM and CML proteins. Methylation, Acetylation, Ubiquitination, and Phosphorylation sites were predicted on the CUCKOO Workgruop and NetPhos 3.1, respectively. For each CaM and CML proteins, the putative Methylation (in blue), Acetylation (in orange), Ubiquitination (in grey), and Phosphorylation (in yellow) sites were shown. The length of colored bars corresponded to the number of expected locations.

**Figure 7 genes-14-01540-f007:**
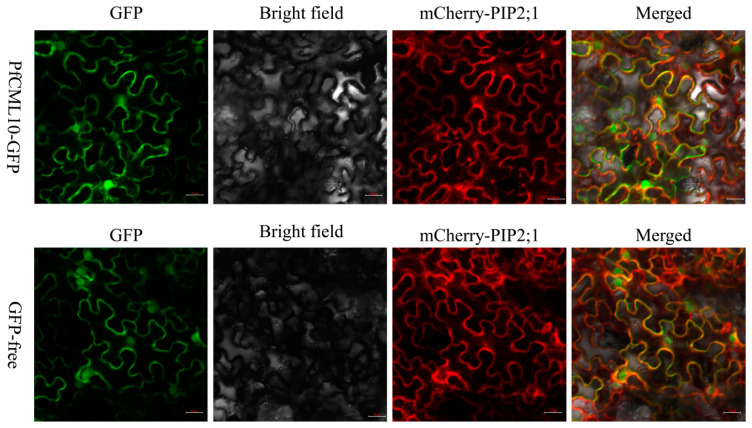
Subcellular localization of PfCML10. *35S:PfCML10-GFP* and *35S:GFP* constructs were individually injected into the epidermal cells of *N*. *benthamiana*. The transient expression of PfCML10-GFP was observed and captured by a confocal laser scanning microscope. Autofluorescence of the GFP-free was also observed and photographed. mCherry-PIP2;1 as PM marker. Scale bars were 20 μm.

**Figure 8 genes-14-01540-f008:**
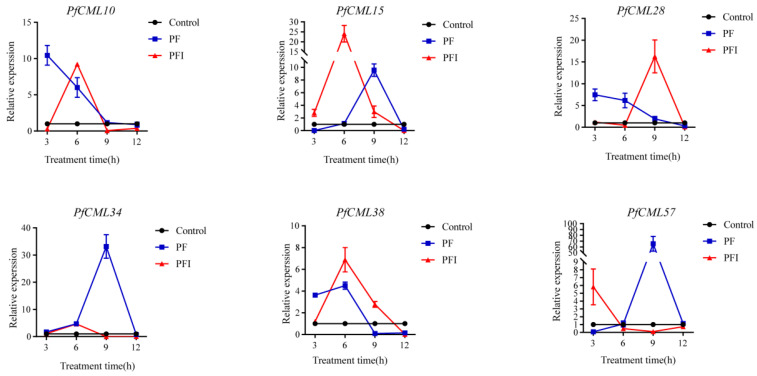
Expression patterns of 6 *PfCMLs* under calcium treatment. Apical buds were collected at 3, 6, 9, and 12 h after treatment. The red and blue lines represented PF and PFI treated with 10 mM CaCl_2_, respectively, and black line represented untreated PF as control. Error bars represented SD values from three biological repeats.

**Table 1 genes-14-01540-t001:** Characteristics of 63 *PfCaM*/*CML* genes in the *Paulownia fortunei*.

Gene Symbol	Gene Locus	Group	PL/(aa) ^(a)^	MW/KDa ^(b)^	pI ^(c)^	C27 ^(d)^	L116 ^(e)^	M ^(f)^
PfCaM1	Pfo01g009450	V	149	16.85	4.11	√	√	6%
PfCaM2	Pfo01g009460	V	149	16.85	4.11	√	√	6%
PfCaM3	Pfo03g002270	V	150	16.93	4.07	√	√	6%
PfCaM4	Pfo12g004780	V	188	21.77	4.8	√	√	5%
PfCaM5	Pfoxxg016130	V	149	1685	4.11	√	√	6%
PfCML1	Pfo02g003340	VI	205	23.00	5.64			5%
PfCML2	Pfo02g016830	VI	185	21.05	6.17			4%
PfCML3	Pfo03g001430	I	186	20.80	4.55			3%
PfCML4	Pfo03g005560	VII	231	25.50	4.8		√	4%
PfCML5	Pfo05g003830	IV	152	17.17	4.03			7%
PfCML6	Pfo05g008280	VII	154	17.40	4.48			6%
PfCML7	Pfo05g014450	II	274	31.04	5.87			2%
PfCML8	Pfo06g000150	III	169	19.46	4.79		√	7%
PfCML9	Pfo06g009190	VI	84	9.22	4.42			5%
PfCML10	Pfo06g009210	VI	151	16.51	4.64			3%
PfCML11	Pfo06g013840	IX	195	22.35	4.88			3%
PfCML12	Pfo07g003100	IX	191	21.72	4.55			2%
PfCML13	Pfo07g004220	III	161	17.82	4.29			6%
PfCML14	Pfo07g006290	VI	189	20.76	4.49		√	4%
PfCML15	Pfo07g008300	III	164	18.07	4.5			2%
PfCML16	Pfo08g007110	VI	167	18.62	4.59			5%
PfCML17	Pfo08g009080	IV	149	16.95	4.15	√	√	6%
PfCML18	Pfo09g010040	II	229	25.93	4.48			5%
PfCML19	Pfo10g000920	IX	138	15.28	4.42	√		7%
PfCML20	Pfo10g005120	IV	147	16.60	4.8			3%
PfCML21	Pfo11g000490	II	157	17.28	4.36			4%
PfCML22	Pfo11g002860	IV	152	17.02	3.97	√	√	5%
PfCML23	Pfo11g005070	II	246	28.74	5.04	√		3%
PfCML24	Pfo11g011500	IX	196	21.85	5.04			3%
PfCML25	Pfo11g014110	VIII	155	17.58	4.62			5%
PfCML26	Pfo11g014120	VIII	158	18.59	4.48			3%
PfCML27	Pfo11g014730	VI	156	16.76	3.67			6%
PfCML28	Pfo12g001970	IV	152	17.02	4.03			6%
PfCML29	Pfo12g011100	VII	153	17.40	4.56			7%
PfCML30	Pfo13g006910	VI	84	9.19	4.52			4%
PfCML31	Pfo13g006920	VI	153	16.48	4.31			4%
PfCML32	Pfo13g008390	VI	179	20.04	6.31			6%
PfCML33	Pfo13g008930	IX	190	21.50	5.23			3%
PfCML34	Pfo13g010140	IX	200	22.98	4.51			3%
PfCML35	Pfo13g011810	IX	141	15.89	4.48		√	6%
PfCML36	Pfo14g004090	IV	151	16.95	4.15	√	√	7%
PfCML37	Pfo14g008380	IX	195	22.16	4.45			5%
PfCML38	Pfo15g003460	VIII	151	17.60	4.48			2%
PfCML39	Pfo15g003470	VIII	159	18.33	4.53			4%
PfCML40	Pfo15g007850	IX	213	24.27	5.71	√		2%
PfCML41	Pfo16g003140	IV	103	11.87	4.15			8%
PfCML42	Pfo16g008660	II	229	26.30	4.57			4%
PfCML43	Pfo17g000060	VI	190	21.18	4.5			2%
PfCML44	Pfo17g003910	IX	193	22.19	4.62			4%
PfCML45	Pfo18g012180	III	164	17.98	4.65			2%
PfCML46	Pfo19g000740	II	159	17.14	4.53			3%
PfCML47	Pfo19g004470	IV	152	17.18	3.97	√	√	6%
PfCML48	Pfo19g009400	VII	222	25.43	5.42			5%
PfCML49	Pfo20g007420	IV	147	16.57	4.79			3%
PfCML50	Pfoxxg000660	VIII	189	21.17	4.45			4%
PfCML51	Pfoxxg003060	II	275	30.53	8.94			1%
PfCML52	Pfoxxg003290	I	268	29.93	4.69			2%
PfCML53	Pfoxxg004030	II	241	26.49	5.72			1%
PfCML54	Pfoxxg004220	I	265	29.48	4.85			2%
PfCML55	Pfoxxg018040	II	258	28.11	5.92			1%
PfCML56	Pfoxxg018260	I	221	24.50	4.48			3%
PfCML57	Pfoxxg024780	VIII	189	21.19	4.45			4%
PfCML58	Pfoxxg028190	I	221	24.50	4.48			3%

^a^ Presence of protein length. ^b^ Presence of molecular weight. ^c^ Isoelectric point prediction by ExpasyProtParam server. ^d^ Presence of a cysteine equivalent to Cys27 of typical plant CaMs at residue 7(-Y) of the first EF-hand. ^e^ Presence of a lysine equivalent to Lys116 of typical plant CaMs. ^f^ Percentage of methionine residues in the deduced amino acid sequence.

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
