# Peer review of "Genome-Wide Characterization of Calmodulin and Calmodulin-like Protein Gene Families in Paulownia fortunei and Identification of Their Potential Involvement in Paulownia Witches’ Broom"

_genes, 2023, doi:10.3390/genes14081540_

Round 1

Reviewer 1 Report

The manuscript "Genome-wide Characterization of Calmodulin and Calmodulin like (CaM/ CML) Gene family in Paulownia fortunei and Identification of the Potential CaM/CMLs Involved in Paulownia Witches' Broom" is well designed.
Here are just a few suggestions:
- Describe the botanical affiliation of the species Paulownia fortunei
- Use Paulownia fortunei rather than P. fortunei in the captions and text below the figure. Also, in each section of the article where the genus Paulownia fortunei is first mentioned, use the full title of the genus.
This will make each part of the manuscript more readable.
The article is scientifically interesting.

Reviewer 2 Report

This manuscript presents a comprehensive genome-wide identification of the Calmodulin and Calmodulin-like (CaM/CML) gene family in Paulownia fortunei. The identification has been executed in a meticulous manner, employing both bioinformatics analysis and molecular experiments to ensure a thorough investigation.

My major comment: 1. This paper shows significant similarities to another previously published paper titled "Genome-wide identification and expression analysis of calmodulin and calmodulin-like genes in apple (Malus × domestica)" (DOI: 10.1016/j.plaphy.2019.04.014) in terms of research content, manuscript structure, and main figures. Certain portions of the Materials and Methods section appear to be identical. However, the authors have not appropriately cited or disclosed this borrowing of content from the other paper within their manuscript. 

2. The method used for the evolutionary analysis is inappropriate. Perhaps influenced by the paper they referenced, the authors employed the neighbor-joining method, while the three species under analysis have a considerable phylogenetic distance. The maximum likelihood method would have been a more suitable choice. Concealing bootstrap values below 50 suggests a lack of confidence in the accuracy of this analysis and an attempt to cover up the shortcomings using shortcuts. I recommend that the authors reconsider the choice of alignment methods and parameters to improve the robustness of the results.

3. The Materials and Methods section does not align with the results section. For instance, Figure 3 depicts the chromosomal distribution and collinearity of the studied genes, but the Materials and Methods section does not mention the methods for analyzing collinearity. Moreover, the term "chromosome density" is problematic, and it is presumed to be "gene density". Likewise, a suitable method description for this is missing.

My minor comments:

1. In the process of selecting candidate genes in P. fortunei, the authors need to provide a detailed description of how they carried out the "manual inspection" for the readers' reference. It would also be ideal if the authors could also provide the list of Arabidopsis genes used for identification.

2. There is an issue regarding the standardization of species names. The authors use the Latin name, P. fortunei, when referring to the species under study. However, when mentioning species from other papers, one is referred to by its common name, "apple," while the other is referred to only by its genus. This does not adhere to the established standards. Please correct this.

3. In section 2.5, the authors should clarify the method used to define "The 2000bp promoter sequences".

4. In section 2.3, the URL for GSDS 2.0 is no longer active. Please update it.

5. The identification of CaM/CML genes in apple should be included in the Materials and Methods section rather than in the legend for Figure 1.

6. The starting points of Figures 1 and 3 should be adjusted. It is recommended to begin from the direction of 12 o'clock on the clock to facilitate reader comprehension.

7. There is a missing space in line 226.

8. In line 231, the word "sequence" is incomplete.

9. The version number of the software, such as TBtools, needs to be specified.

The level of English writing is acceptable and does not hinder comprehension. However, further improvement in this aspect would be preferable.
